# Social Understanding of Disability: Determinants and Levers for Action

**DOI:** 10.3390/bs14090733

**Published:** 2024-08-23

**Authors:** Ulysse Lecomte, Araceli de los Ríos Berjillos, Laetitia Lethielleux, Xavier Deroy, Maryline Thenot

**Affiliations:** 1Social Matters Research Group, Universidad Loyola Andalucía, 14004 Córdoba, Spain; arios@uloyola.es; 2Management Department, Université de Reims Champagne Ardenne, 51571 Reims, France; laetitia.lethielleux@univ-reims.fr; 3Strategy & Organization Studies, NEOMA Business School, 76130 Mont-Saint-Aignan, France; xavierderoy@gmail.com (X.D.); maryline.thenot@neoma-bs.fr (M.T.)

**Keywords:** perception of disability, social psychology, social image, social inclusion, social justice, new generations, levers of action

## Abstract

The prejudices often associated with the perception of people with disability can limit their access to the opportunities and resources available in society, leading them to live in a climate of great socio-economic uncertainty exacerbated since the COVID-19 pandemic. This research focuses on the perceptions of young people in France, defined as those aged between 18 and 30, towards people with disability. The study draws on the principles of social psychology to understand these perceptions, the factors that influence them and the most effective ways of promoting greater inclusion. A survey of 660 young people confirms that, despite recent progress, people with disabilities are still perceived as socially excluded. The results show that familiarity with disability, open-mindedness, the visibility of disability and the quality of interactions with people with disabilities have a strong influence on perceptions. To improve these perceptions, disability training and awareness raising are considered more effective than communication or positive discrimination measures. This research is the first to explore perceptions of disability among young people in France, with the potential to influence future behavior. It suggests ways to promote effective inclusive practices and support policies that encourage positive interactions with people with disabilities.

## 1. Introduction

In France, the welfare state policies of the 20th century led to significant progress for the country’s 12 million people with disability. Initially focused on the medical and individual aspects of disability, they gradually incorporated its social dimension, recognizing that society’s failure to adapt to the needs of people with disability was a cause of disability. In response, collective measures were adopted to facilitate their access to public and private spaces and to promote their inclusion into the world of work, education, culture and sport. However, although these measures have improved the quality of life of people with disabilities, they have not yet produced the expected results leading them to live in a climate of great socio-economic uncertainty exacerbated since the COVID-19 pandemic. People with disabilities have lower average earnings, are more likely to live in poverty, and are stigmatized by a social perception that is often reduced to a consideration of their disability alone [1]. There are significant gaps in employment and social legitimacy for people with disability, due to a persistent reluctance on the part of employers to recruit them, often based on the belief that “disability means low productivity” and, more generally, a reluctance on the part of society to entrust them with responsibility [2]. Furthermore, when confronted with a negative social stereotype, people with disability tend to internalize and conform these thought patterns, in line with the self-fulfilling prophecy phenomenon including in the socio-professional world, as confirmed by [3].

Thus, it appears that social perceptions of disability and a lack of social awareness are the main barriers to an adequate social response and appropriate support for people with disability [4]. The social perception of people with disabilities is, therefore, a major challenge in meeting their needs in terms of improving their inclusion and well-being. However, despite numerous studies defining the concept of the social image of disability [5], none has specifically explored the perceptions of young people in France with regard to disability. Yet today’s young people have a crucial role to play in the social development of people with disability, as they will be tomorrow’s employers, colleagues and policy makers, and their social impact will be all the greater as young people generally have more open and positive attitudes than older people [6,7]. For this reason, we focus on the perceptions of individuals in the 18–30 age group at the time of this exploratory study.

The aim of this study is to examine the social perceptions of young people in France with regard to people with disabilities and to analyze certain factors and levers of action likely to influence them. We hypothesized that these perceptions determine the social image of people with disability and that certain practices and policies could improve their inclusion and quality of life. Our analysis draws on social psychological concepts applied to disability, such as social relations, influence, representation and identity, to examine how people with disability are perceived, treated and integrated into society [8]. Our research stands out for its focus on young people’s social perceptions and sensitivities to disability, two crucial elements that influence their attitudes and future choices, which can either promote or hinder the inclusion of people with disability and their quality of life. It complements the study by [9], which focused exclusively on the perception of intellectual disability by American students in the same age group.

First, we present our literature review, then our methodology based on a questionnaire on the perception of disability, for people aged 18 to 30, living in France, before presenting the results in detail. Finally, we discuss the innovative contribution of our study, which advocates the promotion in France of policies and actions considered legitimate, aimed at improving the perception of disability and enabling people with disability to participate actively in their own social inclusion.

## 2. Literature Review

Research in social psychology [10] on the formation of judgements seems enlightening and adaptable to the perception of disability. Ref. [11] argues that our judgements are strongly influenced by our social representations, past experiences shared knowledge, prejudices and stereotypes associated with the social groups to which we belong. These interpretations mobilize standards of reference to evaluate others, thus influencing our perceptions and generating specific expectations. Psychological congruence often takes precedence over logical coherence, which simplifies reality but can lead to errors in the evaluation of individuals. Therefore, our literature review aims to deepen the understanding of attitudes, beliefs and behaviors toward disability by examining how people with disability are perceived, treated and integrated into society, as highlighted by [8].

### 2.1. The Shaping Social Image as an Element Perception of Disability

Social identity refers to the way in which individuals perceive themselves and are perceived as members of a social group and the social perception of a population plays a fundamental role in its social inclusion or exclusion [12,13,14]. For this reason, a great deal of research has gone into defining the concept of the social image of disability, given its importance in normalizing, integrating and improving the quality of life of this population. This quality of life includes various dimensions such as emotional well-being, interpersonal relationships, personal development, physical well-being, self-determination, social inclusion and access to greater rights [5].

However, people with disabilities are often still perceived as a marginalized minority with a historically weak political position, largely due to stereotypes or prejudices that distort their social image and their real ability to engage and contribute to social life [15]. Thus, based on the Stereotype Content Model [16] enriched by the work of [17] defining social prejudice around three dimensions, warmth, competence and courage, people with disability are generally judged as pleasant and courageous but lacking in competence, particularly when it comes to more visible disabilities [18]. According to the observations of [4], these stereotypes and prejudices persist due to the lack of repeated stimuli to change them which is often exacerbated by the low level of social awareness and sensitization to disability, as highlighted by [19]. Moreover, socially accepted norms have a strong impact on attitudes and behavior toward disability, encouraging individuals to conform to them. According to [20], they thus help to maintain existing hierarchies and power dynamics in society, which in turn, lead to the exclusion of marginalized groups, thereby reinforcing social inequalities and disparities. These authors also point out that many social behaviors are influenced by the norms conveyed by contemporary social media, which place a high value on image and appearance. The social construction of images shapes collective perceptions of physical characteristics, behaviors and the positive and negative connotations associated with them. Appearance thus becomes a criterion for judging a person’s ability to perform different tasks, contributing to a social classification based on physical appearance [21]. Furthermore, the work of [22] shows that people have a simplified perception of people with disabilities, often indiscriminately grouping them in a homogeneous category, whereas the concept of disability is multifaceted. They are then equated with other disadvantaged groups, often stigmatized for their poor social inclusion, and reduced productivity. As a result, the heterogeneity of this group, its difficulties and their consequences are often misunderstood by many people. Ref. [23] confirms that the social image of disability is largely influenced by this widespread ignorance in society, which hinders social interaction with people with disabilities and better mutual understanding. This ignorance hinders the promotion of supportive behaviors, which could act as a driving force to improve the current situation of people with disabilities. Deliberate ignorance of the challenges and needs of people with disabilities hinders recognition of the collective nature of the phenomenon, which can affect any individual or their family at any time.

Consequently, this social representation, largely influenced by a lack of understanding of disability and by the importance attached to the physical appearance of the individual as a criterion for judging his or her abilities, contributes in part to the disabled population’s lack of social power, i.e., their ability to influence the evolution of society and the behavior of its members [24]. This reduced capacity constitutes an obstacle to the commitment of this population to improve their own quality of life, while society, through its norms and structure, generates different situations of “disaffiliation”. Ref. [25] cites the example of people with reduced mobility who, faced with greater obstacles to using public transport, are less likely to take advantage of the opportunities offered by the urban environment.

### 2.2. Towards a Change in the Societal Perception of Disability

Although the social perception of disability still needs to be improved, the authors point to a significant evolution in society’s attitudes towards people with disability, particularly since the 2000s. Ref. [26] observes that part of society is increasingly recognizing their abilities and skills, leading to greater acceptance and recognition of their value and dignity. For [27,28], a prominent example of this evolution is the inclusive dimension built into the Paris 2024 Olympic Games project, which aims to present the disabled community as active and dynamic. For [26,29], this positive trend is due in particular to the current greater recognition of three fundamental principles of disability inclusion. Firstly, to involve all social forces and public opinion as essential actors in accelerating inclusion processes; secondly, to believe that social behavior is modified or stabilized by systems of exchange and communication; and finally, to go beyond any corporatist vision by working with bodies with financial and political power.

However, there is still a certain stigma attached to disability and its complexity and to make further progress, it is crucial to recognize that disability encompasses a wide variety of situations [30]. The challenge is to find the right job opportunities for each individual, considering their education, skills and potential to make a positive contribution despite their disability. Furthermore, ref. [31] stresses the importance of not neglecting invisible disabilities, which are often ignored in discussions on the inclusion of people with disabilities.

### 2.3. Impact of Practices to Improve the Perception and the Inclusion of Disability

However, some experts suggest effective ways to increase social awareness of and incentives for disability. For example, refs. [32,33] emphasize the crucial importance of schools and universities that integrate students with disabilities, stressing that disability awareness and education must begin in childhood to encourage intergroup contacts. Ref. [34] argues that the individual’s family and social environment also play a crucial role, as the values transmitted by the family are essential in raising young people’s awareness of disability issues and shaping their future prospects as adults. On the other hand, refs. [35,36] highlight the power of the media in shaping opinions and attitudes within the population, making it a crucial tool for general disability awareness. What is more, disability issues are now treated by the press in a more normalized way, far removed from the charitable approaches of the past. The impact of the media is also highlighted by [37], although he argues that it is direct contact with people with disabilities that most influences beliefs and attitudes. Moreover, ref. [38] point out that this influence is all the stronger in the case of contact incompatible with a stereotype, i.e., positive contact with disability by a negatively stereotyped group, or negative contact with disability by a positively stereotyped group.

This literature review highlights the importance of social image and perception in the inclusion of people with disabilities. Due to persistent stereotypes and lack of social awareness, this population remains marginalized with no real ability to contribute to social life. Current social media norms reinforce this exclusion by emphasizing physical appearance. However, some authors note a positive evolution in the image of people with disabilities with a growing social appreciation of their abilities and skills and identify several factors to explain these changes in attitudes.

As part of the validation of this literature review, our empirical study has several aims. Firstly, it aims to deepen our understanding of young people’s perceptions of disability in France. Secondly, it aims to assess the relevance of the representations of disability, the influencing factors and the levers for action identified in the literature review, applied to young people in France. At the same time, our study seeks to identify other potentially important factors for the implementation of policies aimed at improving the quality of life of people with disabilities.

## 3. Method

We designed an online ad hoc questionnaire for young adults aged 18 to 30 living in France, excluding minors to avoid any family influence on their perceptions of disability. We retained 660 responses for this questionnaire, distributed over the period from December 2022 to May 2023, after eliminating 42 responses. These eliminations corresponded either to incomplete questionnaires or to questionnaires completed by people who themselves have a disability or who have a disabled relative and were, therefore, likely to present a significant cognitive bias. The final sample of 660 responses was 76% female and 24% male, with an average age of 23 (standard deviation = 2.33), of whom over 78% were students, 12% employees and 10% not in work. The aim is to understand our sample’s perceptions of disability and the factors that influence them, in order to identify levers for action to improve these perceptions. The various items selected stem from the issues raised by the authors mentioned in the literature review.

The preamble facilitates understanding and encourages respondents to participate. The first paragraph defines the objectives of the study and emphasizes the importance of their contribution, while the second paragraph ensures the confidentiality of the survey and provides a safe environment for respondents to answer freely. No instructions were given to participants to focus on a specific type of disability, with the aim of obtaining a perception of disability in all its diversity.

The aim of the questionnaire was to gather detailed data on young people’s current perceptions of disability in France, the main factors influencing these perceptions and possible strategies and actions to improve them. We pre-tested the draft questionnaire with two management science researchers working on disability to validate its relevance and scientific quality We also submitted the questionnaire to a professional in the disability field, administrator of the AGEFIPH, a French government agency promoting the inclusion of people with disability. This double validation enabled us to improve our questionnaire with regard to our research objectives, before submitting it to eight young people with different study levels to test its level of comprehension. The ethical dimension of the questionnaire was approved by a university research ethics committee. The questionnaire was integrated into the Google Forms online platform and participants were able to access the questionnaire via a web link and complete it online. Their responses were automatically stored in a database in the form of an Excel spreadsheet, making it easy to quickly retrieve and sort responses. To maximize the size of our sample, the questionnaire was distributed through posts on the social networking sites Instagram and X (formerly Twitter). In fact, these two social networks are the most popular with young French people in the age bracket targeted by our research, as demonstrated by a study carried out in 2023 by Reech, an influence marketing agency. In all cases, participants were thanked for their contribution to the study. The final stage involved retrieving the Excel spreadsheet generated by the platform and analyzing the data obtained from the questionnaire responses. Absolute and relative frequencies and averages were calculated for each question to facilitate interpretation of the data. The main findings were then presented graphically.

The first part of the questionnaire (questions 1 to 7) explores perceptions of disability, asking respondents to give their opinion on the definition of ‘disabled person’, as well as on social inclusion, forms of discrimination and other specific difficulties faced by people with disability. Additional questions deal with ambiguous interactions between able-bodied people and people with disabilities, as well as personal experiences of relationships with people with disability. The second part (questions 8 to 11) aims to analyze the possible correlation of certain factors with perceptions of disability. On the basis of a list of items with a more or less direct link to disability, respondents are asked to rate the positive and negative impact of each item on their perception of disability. The third part of the questionnaire (question 12) aims to identify effective levers for action to improve young people’s perceptions of disability. Finally, questions 13 to 15 determine the socio-economic profile of the respondents, which is useful for the analysis of the results.

## 4. Results

The results are presented in the same order as the different parts of the questionnaire, from the perception of disability to the different factors influencing it. They are then interpreted in the discussion section.

### 4.1. The “New Generation’s” Perception of Disability

Figure 1 provides information about the “new generation’s” “snapshot” of people with disability, i.e., the first idea that “comes to mind” when disability is mentioned to them.

Figure 2 and Figure 3 below show the evolution of discrimination against people with disabilities (Figure 2) and their inclusion in the main areas of daily life (Figure 3). Finally, Figure 4 provides a current diagnosis of the social inclusion of people with disabilities.

The following three tables examine the current level of inclusion of people with disabilities. Table 1 illustrates the sample’s perception of this inclusion through concrete examples of prejudice against disability. Table 2 shows the intensity of relationships that young people are willing to have with a functionally diverse population. Finally, Table 3 highlights the different social attitudes towards people with disabilities.

In terms of the contemporary perception of disability by the younger generation, analyzed according to the principles of social psychology [10], our results, illustrated in Figure 1, show that the majority of the sample (70%) is based on an immediate perception, often focus on the visible disability and/or the associated technical aids.

This instantaneous perception, although widespread, is considerably removed from the complex reality of disability, as highlighted by [39], who insists on the infinite diversity of disabilities, whether visible or invisible. Thus, our findings confirm that society maintains a simplistic view of disability, confirming previous research by [22] and also supporting the work of [23] who highlights the significant impact of societal ignorance on the social image of disability.

Figure 2 and Figure 3 provide encouraging insights into the possible evolution of this perception over time. Indeed, discrimination, identified by [40] as a major factor of inequality and social exclusion, appears to be decreasing according to more than 75% of respondents (Figure 2). Furthermore, the level of inclusion of people with disabilities in different aspects of daily life seems to be stable in some areas (leisure and work) and improving in others (transport, public places, schools), with no signs of regression (Figure 3). This positive trend confirms the conclusions of [29,41], who observe a growing awareness of “minority groups” in society.

However, despite this perceived improvement, the progress made does not seem sufficient. The data in Figure 4 show that around 73% of the sample still feel that people with disability are marginalized and 15% feel that they are completely excluded. In addition, the information in Table 1, Table 2 and Table 3 shows that many situations still lead to their social exclusion. These results confirm the findings of [42] on the widespread marginalization of people with disabilities and thus highlight the need for action to increase their social inclusion and consideration.

In terms of prejudice against disability, Table 1 highlights the persistence of certain ‘beliefs’ with negative connotations. These beliefs contribute to portraying people with disabilities as passive, both personally and professionally. The main stereotypes, in descending order of popularity, are as follows: 1—caring for a disabled person takes courage; 2—people with a disability often find it difficult to work; 3—people with a disability need help; 4—it is more difficult for people with a disability to live alone; 5—people with a disability have little or no sex life.

Table 2 looks in more detail at the issue of ‘private’ life, particularly in terms of the intensity of the relationship that respondents are prepared to have with a disabled person. It highlights the fact that light or occasional interactions (such as friendly exchanges, outings or professional interactions during job interviews or at work) are generally well accepted, but that deeper or regular relationships elicit greater reluctance. This is the case for commitments such as starting a family, maintaining a romantic relationship and, to a lesser extent, living with a disabled person. Thus, although they are open to sharing time with people with disabilities, respondents find it difficult to make a long-term commitment and to integrate disability and its specific needs into their lives, as [43] notes.

The data in Table 3 highlight the barriers to the inclusion and relationships of people with disability within society. For example, the majority of respondents do not always know how to help a disabled person, confirming earlier findings about society’s lack of awareness of disability. In addition, these findings highlight a lack of spontaneity and openness among most participants when interacting with people with disabilities, as well as a reluctance to discuss disability. According to [4], this attitude, described as ‘politically correct’, stems from a historical legacy in which disability was a taboo and hidden subject, a mentality that needs to be overcome today. This situation hinders communication between disabled and non-people with disability and thus negatively affects the perception of disability and the social inclusion of all people with disability.

In conclusion, although progress has been made in the current perception of disability by the ‘new generation’, this is still not enough to ensure real social inclusion. This finding corroborates the conclusions of [44], who point out that despite the efforts made to address disability issues, people with disabilities continue to lack social power and remain in a situation of social vulnerability. Various social factors, whether environmental, cultural, or economic, persist and exacerbate this situation.

### 4.2. Factors That May Influence Young People’s Perceptions of Disability

However, the study also explores the factors that are likely to influence these perceptions. Table 4 lists 23 potential influencing factors suggested to respondents and assesses the positive and negative impact of these different factors on their perceptions of disability. This is not an exhaustive list and does not exclude important elements. To make the results easier to understand, the factors examined are numbered and grouped as follows:-Items 1 to 3: Familiarity with the world of disability-Items 4 to 9: Socio-political-economic factors-Items 10 to 12: Cultural factors-Items 13 and 14: Visibility of disability and/or assistive devices-Items 15 to 23: Factors specific to the disabled person

In addition to the initial findings on perceptions of disability, the data in Table 4 explore the factors influencing perceptions of disability, with each participant asked to rate the positive and negative impact of each of the proposed factors. The analysis shows that the respondent’s level of ‘familiarity’ with the world of disability is a very positive influencing factor (items 1, 2 and to a lesser extent 3). These findings support the work of [9,45], suggesting that perceptions are more easily constructed on the basis of structured knowledge about topics that are familiar to the individual.

However, other more general factors, such as those related to the participants’ culture, have a significant impact on their perceptions of disability (factors 10, 11 and 12). It appears that exposure to new cultures through travel or living abroad promotes positive perceptions of disability (factors 10 and 11), whereas resistance to change (factor 12) is identified as having a negative impact. Thus, attitudes that favor novelty, originality and innovation, as well as high levels of open-mindedness, promote greater consideration for people with disability, supporting the findings of [46], a study in Australia. A societal shift towards such attitudes is a step towards improving the situation of people with disabilities who, according to [47], are still too often perceived as an undervalued and marginalized ‘minority’.

On the other hand, the analysis of socio-political-economic factors (items 4 to 9) shows that their influence, whether positive or negative, on the perception of disability is not significant. Contrary to certain preconceptions, which sometimes associate ‘left-wing’ political orientations with a more positive view of people with disabilities and their social inclusion, it appears that respondents’ political convictions have no significant impact on their perceptions of disability. Similarly, the socio-economic level of origin (modest or affluent) and the rural or urban origin of individuals have no significant effect.

Regarding the impact of the level of visibility of the disability itself and/or the use of associated equipment such as wheelchairs or walking sticks (items 13 and 14), the survey results are surprising. Contrary to what might be expected, a high level of visibility seems to be associated with a positive perception of disability on the part of respondents. This observation can be illuminated by the research of [31], who suggest that society tends to categorize individuals, even unconsciously. In the collective imagination, disability is often associated with a visual dimension and visible characteristics. In the presence of an invisible disability, identity markers are blurred, which can disrupt perceptions. Thus, when a non-disabled person observes behaviour or attitudes that they consider ‘socially abnormal’ in a person with an invisible disability, they may not know how to interpret them. This lack of clarity can increase the risk of negative perceptions of the disabled person and, as a result, their social marginalization.

It is also worth noting that the attitudes of people with disability themselves (items 15 to 23) have a significant impact, whether positive or negative, on the way in which the community perceives disability. The results suggest that certain specific characteristics of people with disabilities play an important role in the way they are perceived by society. Among the characteristics that favour a positive perception of people with a disability are their constructive and positive attitude, their communication skills, their charisma, their dynamism, their extroversion and their athletic and sporting abilities. On the other hand, characteristics associated with negative perceptions include aggressiveness, negative attitudes and communication difficulties.

These findings highlight the active role that people with disabilities play in shaping the way they are perceived by non-people with disability and underline the importance of promoting a positive image of their community. This includes an optimistic attitude in the face of challenges, a proactive approach, and a desire to connect with others and share their story and individuality by expressing their emotions, whether verbally or non-verbally in the case of communication difficulties. This type of behaviour can promote the emergence of positive affect in the perceiver by familiarizing him or her with people with disabilities, thus allowing for more positive interactions [48]. On the other hand, a pessimistic or aggressive attitude on the part of a disabled person, especially in response to a feeling of rejection by society because of their ‘functional diversity’, is seen as a negative factor. Indeed, according to [42], although understandable in the face of the frustration of feeling socially excluded, such a negative attitude would only increase the already high risk of social isolation that people with disability have faced in recent decades.

Ultimately, this in-depth exploration of the various factors that have a significant impact, whether positive or negative, on the perception of disability is an essential step in identifying potential levers for action aimed at promoting a better perception and social inclusion of people with disabilities.

The study then looked at ways to improve these perceptions.

### 4.3. Effective Levers for Action to Improve the Perception of Disability

Table 5 presents data on the importance attached to the different levers, which are numbered and grouped as follows:-Levers 1 to 5: Training of stakeholders, including people with disabilities.-Levers 6 to 9: Disability-related communication and media actions.-Levers 10 to 12: Positive discrimination measures.

In our study, we asked our sample for their opinion on the effectiveness of 12 levers to improve the perception of disability. The results are presented in Table 5. It can be seen (items 10 to 12) that the introduction or strengthening of positive discrimination mechanisms in different areas such as work, media, and politics, providing for minimum quotas of representation and/or recruitment of people with disability were identified as potential levers. These mechanisms are based on the principles of fairness and equality opportunity and aim to favor certain social groups that are considered more disadvantaged, to put them on an equal footing with the rest of society, in line with the work of [49].

However, the data collected show that this process is considered to be the least effective by our sample. These results support the thesis of [50] that positive discrimination mechanisms, although intended to correct inequalities affecting certain social groups, may in fact create new inequalities within other groups. Thus, without an appropriate approach, this type of mechanism may simply displace injustice rather than solve it.

Other levers related to communication and media actions in favor of people with disabilities (points 6 to 9) seem to be somewhat more effective. They can take various forms, such as the broadcasting of short films, the implementation of communication campaigns or even the simultaneous organization of the Paralympic and Olympic Games. These observations are in line with the conclusions of [35] who highlight the potential of the media to express opinions and influence attitudes within the population as an effective tool for raising awareness of disability in society. However, it is surprising that, given the young age of the participants, the action lever (item 8), which aims to select people with disability to become influencers and/or communicate on social networks was not considered a priority. This observation could be attributed to the lack of awareness in society about the professional and creative skills of people with disabilities.

In the end, training (items 1 to 5) emerged as the most effective lever (rated more than 4 on a scale of 5) to positively influence perceptions of disability and promote inclusion. This includes disability training for all stakeholders in society, as well as training for people with disabilities themselves to improve their communication skills. These findings support the work of [37], in highlighting the fundamental importance of direct contact with individuals, particularly through training and awareness programs, in shaping beliefs and attitudes.

## 5. Discussion and Conclusions

For more than 20 years, French public policy has considered the social dimension of disability, recognizing the inadequacy of the organization of society to meet the specific needs of 12 million people. Today, the inclusion of people with disabilities is perceived as a major social challenge, reinforced by the growing interest in Corporate Social Responsibility (CSR) and diversity. However, despite the introduction of infrastructure and services to improve accessibility in everyday life, people with disabilities remain socially and professionally marginalized. Several studies highlight the strong impact of social perceptions on this exclusion but are limited to the case of intellectual disability in North American countries [6,9]. Our study makes a significant contribution to the existing literature by addressing two aspects: firstly, it explores the perception of disability in all its diversity among young people in France, and secondly, it highlights the levers of action they consider legitimate to promote the inclusion of people with disability. It also analyzes the impact of the behaviour of people with disability on their own image. Overall, the study replicates some of the findings of [6] and shows that in France, as in North American countries, the perception of disability is evolving positively, with less discrimination and more inclusion in the different aspects of daily life, which offers hope for the future. This evolution is the result of the first effects of the French law n° 2005-102 of 2 November 2005, which sets 2015 as the target date for the mandatory accessibility of the entire transport chain (buildings, roads, public transport, etc.), and in particular access for people with disability to education and employment, thereby increasing their frequency of interactions with society. However, the study also confirms that this progress is not enough. In 2023, for example, the French National Conference on Disability noted that the objectives of the 2005 law had only been partially achieved, thus undermining inclusion. Many stereotypes persist, particularly related to a lack of awareness of disability and its complexities. For example, most of the young people who responded to the questionnaire feel unease and pity towards people with disability, lack spontaneity in their interactions, and may be afraid to talk about disability. Yet, such strong feelings can be inappropriate and devaluing for people with disabilities who, if they internalize these negative perceptions, risk voluntarily excluding themselves from society. What is more, respondents often perceive people with disability as passive, reflecting a lack of awareness of their true abilities. They are also reluctant to develop close, long-term private relationships with them, not least because of the complexity of dealing with disability, a behaviour that may be influenced by the individualistic nature of our societies.

Consequently, these findings, as suggested by other studies [6,7], highlight the growing need for awareness campaigns and educational programs aimed at promoting interaction with people with disabilities. Indeed, increased contact with these individuals helps to overcome the negative and discriminatory barriers that prevent the recognition of their positive characteristics [51]. Thus, by providing early and regular contact, mainstream education for children with disabilities helps to improve the image of disability. In France, since 2013, the law requires the public service to ensure the educational inclusion of all children, without distinction, and to offer them vocational training opportunities to facilitate their future professional integration. This educational inclusion is complex, as it requires adapted educational practices and materials, as well as teachers trained to meet the specific needs of students with disabilities while respecting the principle of common education [52]. In addition, our results show the strong influence of the media across all categories and of major events, such as the simultaneous organization of the Olympic and Paralympic Games in France in 2024, on perceptions of disability [27,28]. However, training is undoubtedly the most effective means of promoting a better understanding of the diversity and complexity of disability in order to shape more positive beliefs and attitudes. Furthermore, encouraging such training to be delivered by people with disabilities would promote quality interactions and counter certain stereotypes about their professional abilities. Another interesting aspect of this study is that it highlights the importance of training for people with disabilities to improve their communication skills, increase their self-confidence and their self-image, and thus avoid social isolation [53]. In this way, they can better accept their disability and play an active role in their successful social inclusion. However, this success depends on the extent to which they are exposed to overly negative social perceptions, which are particularly exacerbated when they come from people with little knowledge of and little interaction with disability.

Thus, despite the contextual nature of perceptions of disability, our research suggests coherent ways of promoting initiatives in favor of inclusive practices. The results underline the importance of promoting policies aimed at improving perceptions of disability and enabling people with disability to actively participate in their own social inclusion.

Regarding the limitations of this research, it should be noted that all were volunteers to answer the questionnaire (informed consent). We can, therefore, assume that these people are already aware of, or at least interested in, disability issues. In addition, the fact that the sample was largely composed of women and students may have an impact on generalizability. Finally, the influencing factors and levers for action on the perception of disability highlighted in this article reflect the responses of our sample of young non-experts about disability.

These results could form the basis of a second study that is more focused on intergenerational comparison of the perception of disability. We also plan to use them for further research into the issue of disability in the workplace in France, where the minimum employment rate of 6% is often unattainable, even in some of the largest CAC 40 companies.

## Figures and Tables

**Figure 1 behavsci-14-00733-f001:**
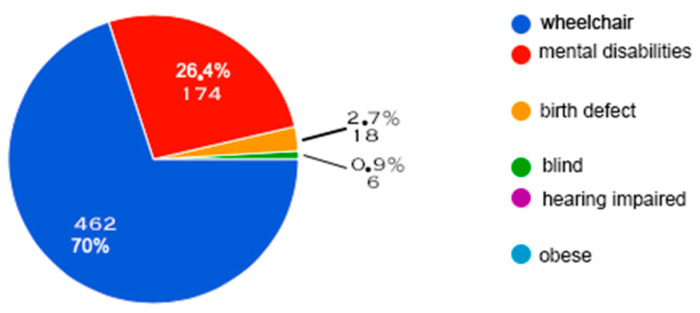
Snapshot of the new generation’s perception of disability.

**Figure 2 behavsci-14-00733-f002:**
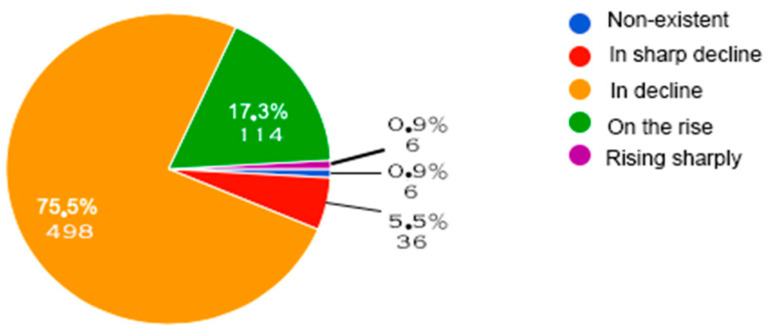
Changes in discrimination against people with disabilities based on responses from the New Generation sample.

**Figure 3 behavsci-14-00733-f003:**
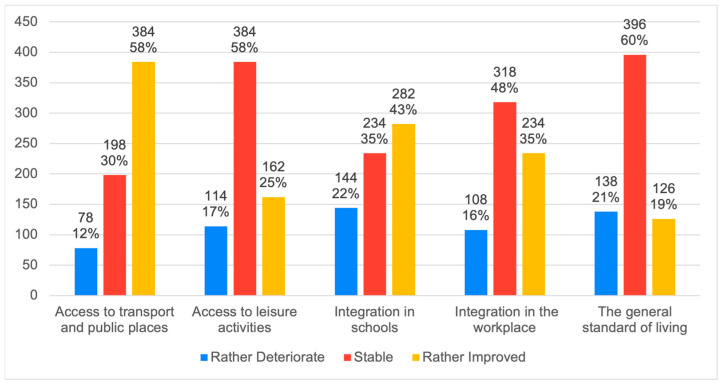
Changes in the inclusion of people with disabilities in the main living spaces of everyday life according to the New Generation sample.

**Figure 4 behavsci-14-00733-f004:**
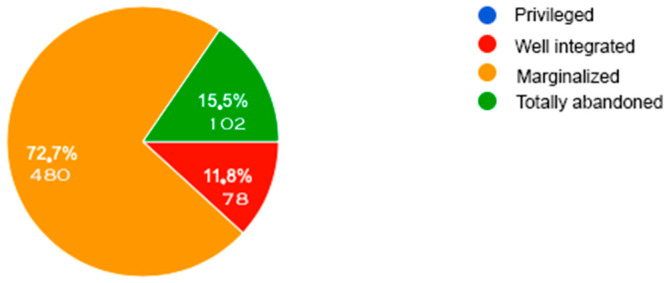
Current levels of social inclusion of people with disabilities according to our New Generation sample.

**Table 1 behavsci-14-00733-t001:** Preconceived ideas about people with disabilities according to the “new generation” sample (Likert scale from 1 to 5, with 1—not at all in agreement and 5—strongly in agreement).

Item	M	SD
Caring for a person with a disability takes courage	4.14	0.93
Disability in general is costly to society	3.07	1.19
If science progressed faster, there would be fewer disabilities	3.28	1.31
People with disabilities find it difficult to live alone	3.75	1.08
People with disabilities generally need help	3.79	0.91
It is more difficult for a disabled person to be fulfilled in his social life	3.55	1.08
It is more difficult for a disabled person to work	3.94	0.97
People with disabilities are more self-contained	2.62	1.01
People with disabilities have no/low sex life	3.45	1.01
People with disabilities are “role models”	3.10	1.26
People with disabilities sometimes take advantage of the benefits reserved for them	2.14	1.17
Sometimes people with disabilities must want to die	2.71	1.03

**Table 2 behavsci-14-00733-t002:** Ability to relate to a person with a disability according to the intensity of the relationship according to the “new generation” sample (Likert scale from 1 to 5 with 1—very easy and 5—very difficult).

Item	M	SD
Having a friendly relationship with a person with a disability	4.70	0.67
Working with a disabled person	4.85	0.43
Hiring a person with a disability	4.64	0.65
Going to the cinema/restaurant/going out with a person with a disability	4.73	0.59
Living with a disabled person	3.95	1.14
Being in a romantic relationship with a person with a disability	3.10	1.26
Starting a family with a disabled person	2.98	1.35

**Table 3 behavsci-14-00733-t003:** Societal behavior towards people with a disability according to the “new generation” sample (Likert scale from 1 to 5, with 1—not at all in agreement and 5—strongly in agreement).

Item	M	SD
We are sometimes reluctant to talk to a person with a disability about his or her disability	3.98	0.87
We do not always know how to help a person with a disability	4.12	0.94
There is a lack of spontaneity and sincerity towards people with disabilities	3.76	1.04
There is sometimes a feeling of pity towards people with disabilities	2.44	1.06

**Table 4 behavsci-14-00733-t004:** Positive and negative influence of factors on “the new generation’s” perception of disability (Likert scale from 1 to 5, with 1—no influence and 5—very great influence).

Item	Positive Influence		Negative Influence	
	M	SD	M	SD
1. Having knowledge about disability	4.65	0.67	1.45	1.04
2. Visiting a person with a disability	4.64	0.60	1.46	1.06
3. Getting to know famous people with disability through the media	3.64	1.16	1.76	1.20
4. Being politically ‘left-wing’.	1.73	1.15	1.62	1.11
5. Being politically “right-wing	1.36	0.66	2.00	1.37
6. Having a modest social background	1.96	0.07	1.76	1.12
7. From a well-to-do social background	1.85	0.13	1.95	1.22
8. From a rural origin	2.01	1.11	1.83	1.10
9. From an urban origin	2.09	1.18	1.67	1.03
10. Country/culture of origin	3.15	1.22	2.23	1.18
11. Having lived in countries with different cultures	3.31	1.24	1.69	1.07
12. Resistance to progress	1.72	1.08	3.28	1.59
13. Degree of visibility of disability	3.94	1.06	2.21	1.27
14. Presence of wheelchair or other visible material aid	3.25	1.11	2.01	1.19
15. The ease with which the person with a disability can communicate	4.06	1.05	1.54	1.04
16. The difficulty of the disabled person to communicate	2.28	1.24	2.94	1.24
17. The positive attitude of the disabled person towards his/her disability	4.32	0.98	1.54	1.02
18. The negative attitude of the disabled person towards his/her disability	2.03	1.06	3.25	1.21
19. Extroverted behaviour of the disabled person	3.71	1.14	1.68	1.06
20. The aggressiveness of the disabled person	1.79	1.10	3.48	1.22
21. The dynamism of the disabled person	3.91	1.10	1.56	1.00
22. The charisma/Leadership of the disabled person	3.93	1.10	1.58	1.00
23. Athletic and sporting abilities of the person with a disability	3.68	1.12	1.62	1.03

**Table 5 behavsci-14-00733-t005:** Levers of action and their perceived effectiveness in improving the perception of disability. according to “the new generation” sample. (Likert scale from 1 to 5, with 1—not effective and 5—very effective).

Item	M	SD
1. Impose mandatory disability awareness training in all companies	4.15	0.97
2. Train people with disability to communicate better about their disability	4.20	0.94
3. Require teacher training in the reception and inclusion of young people with disability	4.47	0.83
4. Training and awareness-raising of students on welcoming and inclusion of people with disability	4.35	0.95
5. Awareness raising and providing training for the inclusion of parasports sections in non-disabled person clubs	4.13	1.06
6. Organizing Olympic and Paralympic Games events during the same period	3.75	1.17
7. Production of anti-prejudice short films featuring active people with disability	3.74	1.19
8. Selecting people with disability to become influencers on social networks	3.31	1.35
9. Develop a humorous communication campaign to make disability less taboo	3.52	1.29
10. Increase the percentage of mandatory recruitment of people with disability in companies and organizations	3.40	1.24
11. Set a minimum percentage of representation of people with disability in the media	3.32	1.18
12. Impose a minimum percentage of people with disability in the political landscape	3.27	1.31

## Data Availability

The data presented in this study are available on request from the corresponding author.

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
