# Peer review of "Social Understanding of Disability: Determinants and Levers for Action"

_behavsci, 2024, doi:10.3390/bs14090733_

Round 1

Reviewer 1 Report

Comments and Suggestions for Authors

This article looks at perceptions of disability among French young people. There is a need for more research in this area, but I have a number of substantial concerns:

Literature Review

The literature review is well written in style but does not engage strongly enough with relevant and recent social psychological research and theory. For example, a reference from 1948 is used to illustrate that disabled people self-stereotype. But a 2009 study (Louvet et al., Swiss J. Psychol.) looked at the extent that French disabled people conformed to stereotypes in work-related contexts.

The literature review mentions several different social psychological concepts (e.g., stereotypes, social identity, intergroup contact, prejudice) but does not explicitly discuss the theories these concepts are based on. For instance, there is a large literature that groups are stereotyped based on two dimensions (stereotype content model) and that disabled people are stereotyped as warm, but incompetent. The existence of these stereotypes has implications for the effectiveness of prejudice reduction interventions like intergroup contact. E.g., Zingora et al. find that stereotype-inconsistent contact predicted changes in attitudes better than stereotype-consistent contact. Carew et al. 2019 looked at the extent media-based contact could shift the stereotypes participants held about disability. These are examples of some relevant studies.

In relation and due in part to the above, it is also not clear to me how the study is building on past research. Yes, a deeper understanding of young people's perceptions of disability in France is needed. But  assessing “relevance in our context of the representations of disability, the influencing factors and the levers for action.”…This is not clear to me as several of the concepts mentioned have strong existing empirical evidence (e.g., intergroup contact and prejudice). Also, I think the review neglects some recent social psychological evidence both in the area in general (see above) and in France specifically (suggest looking at the work of Odile Rohmer like:

Granjon, M., Rohmer, O., Popa-Roch, M., Aubé, B., & Sanrey, C. (2023). Disability stereotyping is shaped by stigma characteristics. Group Processes & Intergroup Relations, 13684302231208534.

Method and Results:

Each of the concepts mentioned in the literature review (e.g., stereotypes, prejudice, contact) etc., have established measures and scale items to assess them. But in my understanding, the authors have generated a scale to look at each of these concepts. However, there is no approach to scale validation described apart from consultation with two management science researchers, which is not adequate enough to convince me that the authors are measuring what they want to measure.

In the case of prejudice (Table 1) it is not clear to me what each of the five scale points correspond to or even if the items tapped respondent’s own personal prejudices or the prejudices they believe French society to hold. Also, these items are very cognitive, but many theories of prejudice also posit that it has an affective and behavioural dimension too (e.g., BIAS map; Fiske et al.) For instance, affective prejudice would be measured by asking what emotions participants felt when interacting with members of the other group. Certainly, the measure used here captures neither affective or behavioural prejudice. This links to my point above about the need to engage in depth with contemporary social psychological theory so the authors can be clear about what they aimed to measure.   

The aim of assessing “relevance in our context of the representations of disability, the influencing factors and the levers for action” also raised my expectation that the research would assess the correlation between various influencing factors and key outcomes like prejudice. E.g., “visiting persons with disability” is similar to contact (Table 4, Item 2). But in actuality the study has not assessed influencing factors. It has assessed what young French people say might be influencing factors. And- unfortunately and sorry to say- I think this is of limited empirical relevance, since the sample is unlikely to be able to accurately assess what can shift their (or similar others') perceptions of disability. It would have been better to measure these different things directly (e.g., contact, political orientation, urban/rural origin) and look at the degree they could predict key outcomes among the sample.

Lastly, there is not enough detail about the sample. What were there demographics and degree of existing experience and contact with disabled people. Were any disabled? All these factors, would influence the responses given.

I do not believe the issues I pointed out are redressable with a revision. Hence I stop the review here.

Minor

p. 2 line 93- presupposes reader is from Western society. Negative preconception about disabled people applies to non-Western societies too. See e.g., Rohwerder 2018 Disability stigma in developing countries

Comments on the Quality of English Language

Good quality. Minor editing required

Author Response

Thank you very much for taking the time to review our research manuscript and for your comments, which have been the basis for the improvement of the study. Please find the detailed responses below and the corresponding revisions in the re-submitted files.

REVIEWER 1:

Comments 1: The literature review is well written in style but does not engage strongly enough with relevant and recent social psychological research and theory. For example, a reference from 1948 is used to illustrate that disabled people self-stereotype. But a 2009 study (Louvet et al., Swiss J. Psychol.) looked at the extent that French disabled people conformed to stereotypes in work-related contexts.

Response 1: We have updated the 1948 reference with the addition of the article by Louvet et al (2009), whose study is perfectly suited to the context of our study (page 1, line 44).

Comments 2: The literature review mentions several different social psychological concepts (e.g., stereotypes, social identity, intergroup contact, prejudice) but does not explicitly discuss the theories these concepts are based on. For instance, there is a large literature that groups are stereotyped based on two dimensions (stereotype content model) and that disabled people are stereotyped as warm, but incompetent. The existence of these stereotypes has implications for the effectiveness of prejudice reduction interventions like intergroup contact. E.g., Zingora et al. find that stereotype-inconsistent contact predicted changes in attitudes better than stereotype-consistent contact. Carew et al. 2019 looked at the extent media-based contact could shift the stereotypes participants held about disability. These are examples of some relevant studies. In relation and due in part to the above, it is also not clear to me how the study is building on past research. Yes, a deeper understanding of young people's perceptions of disability in France is needed. But assessing “relevance in our context of the representations of disability, the influencing factors and the levers for action.”…This is not clear to me as several of the concepts mentioned have strong existing empirical evidence (e.g., intergroup contact and prejudice). Also, I think the review neglects some recent social psychological evidence both in the area in general (see above) and in France specifically (suggest looking at the work of Odile Rohmer like: Granjon, M., Rohmer, O., Popa-Roch, M., Aubé, B., & Sanrey, C. (2023). Disability stereotyping is shaped by stigma characteristics. Group Processes & Intergroup Relations, 13684302231208534.

We have strengthened our literature on the theories underpinning the various concepts of social psychology. We drew on work related to the notions of stereotype (page 3, line 97: Fiske et al. (2007), Rohmer & Louvet (2011), Granjon et al. (2023)), intergroup contact (page 4, line 168: Zingora et al. (2020)) and media-based contact (page 3 line 140: Carew et al. (2019)).

Comments 3: Each of the concepts mentioned in the literature review (e.g., stereotypes, prejudice, contact) etc., have established measures and scale items to assess them. But in my understanding, the authors have generated a scale to look at each of these concepts. However, there is no approach to scale validation described apart from consultation with two management science researchers, which is not adequate enough to convince me that the authors are measuring what they want to measure.

With regard to validation, the process was more detailed: in addition to the validation by 2 academic experts, we specified that the questionnaire was also submitted and validated by a professional in the field of disability, who is also an administrator of AGEFIPH, the French governmental reference agency for promoting the integration of people with disabilities. A pre-test was also carried out with 8 young people corresponding to our sample.

Comments 4: In the case of prejudice (Table 1) it is not clear to me what each of the five scale points correspond to or even if the items tapped respondent’s own personal prejudices or the prejudices they believe French society to hold. Also, these items are very cognitive, but many theories of prejudice also posit that it has an affective and behavioural dimension too (e.g., BIAS map; Fiske et al.) For instance, affective prejudice would be measured by asking what emotions participants felt when interacting with members of the other group. Certainly, the measure used here captures neither affective or behavioural prejudice. This links to my point above about the need to engage in depth with contemporary social psychological theory so the authors can be clear about what they aimed to measure.  

For each table (from 1 to 5), we have indicated the points of each Likert scale used. To answer the reviewer's question about the items in table 1, we would point out that the questionnaire asks respondents for their opinion on potential prejudices from a personal point of view.

Comments 5: The aim of assessing “relevance in our context of the representations of disability, the influencing factors and the levers for action” also raised my expectation that the research would assess the correlation between various influencing factors and key outcomes like prejudice. E.g., “visiting persons with disability” is similar to contact (Table 4, Item 2). But in actuality the study has not assessed influencing factors. It has assessed what young French people say might be influencing factors. And- unfortunately and sorry to say- I think this is of limited empirical relevance, since the sample is unlikely to be able to accurately assess what can shift their (or similar others') perceptions of disability. It would have been better to measure these different things directly (e.g., contact, political orientation, urban/rural origin) and look at the degree they could predict key outcomes among the sample.

In the limitations of the study, it is specified that the sample is not made up of disability experts (page 15, line 513). However, the interest of the study lies in knowing the perception of disability by the young population in France, which has an important potential influence on the evolution of future behaviors.

Comments 6: Lastly, there is not enough detail about the sample. What were there demographics and degree of existing experience and contact with disabled people. Were any disabled? All these factors, would influence the responses given.

We have added details concerning the sample in the "method" section. We have clarified important demographic information (age, gender, socio-professional category) as well as the exclusion of responses from people with disability or their relatives, who present a high risk of cognitive bias (page 4, line 188).

Comments 7: p. 2 line 93- presupposes reader is from Western society. Negative preconception about disabled people applies to non-Western societies too. See e.g., Rohwerder 2018 Disability stigma in developing countries

We have removed the reference to western society to take account of the widespread phenomenon of negative preconception about people with disability (page 2, line 94).

Reviewer 2 Report

Comments and Suggestions for Authors

The paper is well contextualized. Provides a solid review of the evolution of the disability and its current status.

Despite the design of the questionnaire being described, there is a lack of information about the validation process. It is necessary to highlight the research question/main goal and the hypotheses/specific questions need to be more clear.

The results are clear and concise, although somewhat simple. The graphs and tables need to be improved in terms of visualization and understanding.

Discussions align with results. However, it could be improved by trying to delve deeper into everything related to educational and work practices.

The conclusions are well written.

Author Response

REVIEWER 2

Thank you very much for taking the time to review our research manuscript and for your comments, which have been the basis for the improvement of the study. Please find the detailed responses below and the corresponding revisions in the re-submitted files.

Comments 1: The paper is well contextualized. Provides a solid review of the evolution of the disability and its current status. Despite the design of the questionnaire being described, there is a lack of information about the validation process. It is necessary to highlight the research question/main goal and the hypotheses/specific questions need to be more clear.

With regard to validation, the process has been further detailed: in addition to validation by 2 university experts, we have specified that the questionnaire was also submitted to and validated by a professional in the field of disability, who is also an administrator of AGEFIPH, the French governmental reference agency for promoting the integration of people with disabilities A pre-test was also carried out with 8 young people corresponding to our sample (page 5, line 206). The aim of this study was highlighted and the hypotheses clarified: "The aim of this study is to examine the social perceptions of young people in France with regard to people with disabilities, and to analyze certain factors and levers of action likely to influence them. We hypothesized that these perceptions determine the social image of people with disability, and that certain practices and policies could improve their inclusion and quality of life." (page 2, line 56).

Comments 2: The results are clear and concise, although somewhat simple. The graphs and tables need to be improved in terms of visualization and understanding.

For better understanding, the Likert scale points have been specified in the tables, and for each figure we have specified the results both as a value and a percentage. For visualization purposes, the resolution of the figures has been doubled from 300 dpi to 600 dpi.

Comments 3: Discussions align with results. However, it could be improved by trying to delve deeper into everything related to educational practices. The conclusions are well written.

Based on the work of Ebersold et al. (2016), further details have been provided concerning educational practices and their non-negligible impact in improving the image of disability: “Thus, by providing early and regular contact, mainstream education for children with disability helps to improve the image of disability. In France, since 2013, the law requires the public service to ensure the educational inclusion of all children, without distinction, and to offer them vocational training opportunities to facilitate their future professional integration. This educational inclusion is complex, as it requires adapted educational practices and materials, as well as teachers trained to meet the specific needs of students with disability, while respecting the principle of common education [52]”. (page 14, line 488).

Reviewer 3 Report

Comments and Suggestions for Authors

Please see the attached comments.

Comments on the Quality of English Language

English language use was generally fine.

Author Response

REVIEWER 3

Thank you very much for taking the time to review our research manuscript and for your comments, which have been the basis for the improvement of the study. Please find the detailed responses below and the corresponding revisions in the re-submitted files.

Comments 1: The authors refer both to “people with disability” and disabled people”. I would recommend avoiding usingthe latter to ensure that more appropriate and person-centred terminology is consistently used.

As recommended, we have standardized the document with the term "people with disability".

Comments 2: There was some lack of information in “Methods” about the survey, including the overall response rate, response rates for individual items, whether any incomplete surveys were discarded, and so forth.

We have added details concerning the sample in the "method" section. We have specified important demographic information (age, gender, socio-professional category), as well as the non-inclusion of incomplete questionnaires or responses from people with disability or their relatives presenting a high risk of cognitive bias (page 4, line 188).

Comments 3: The presentation of results could be much clearer. For example, while some percentages are provided in the initial figures, there are no corresponding numerical figures. Further, while percentage figures are given for some categories, they are not given for others.

For each figure, we have specified the results in both value and percentage terms, for the sake of harmonization.

Comments 4: Figure 3 contains an item referring to “integration in the company”. What does this mean ?

This item seeks to understand the extent to which the inclusion of people with disabilities in the workplace tends to evolve according to our sample. Thus, replacing the term "company" by "workplace" seems more appropriate, as employment is not only concentrated within companies (page 6).

Comments 5: The Results on page 5 refers to the Cronbach alphas for the scales, but no further information is providedabout these scales and it is not clear what scales were constructed, why, or for what purpose, given that only descriptive statistics are presented. The authors should explain what the response categories were for the survey items. It is noted that a Likert scale from 1 to 5 was used, but no category responses were provided.

The Likert scale points were specified for each table to better understand the usefulness of each scale in this investigation:

Table 1. Preconceived ideas about people with disability according to the "new generation" sample (Likert scale from 1 to 5, with 1-Not at all in agreement and 5-Strongly in agreement).

Table 2. Ability to relate to a person with a disability according to the intensity of the relationship according to the "new generation" sample (Likert scale from 1 to 5 with 1-Very easy and 5-Very difficult).

Table 3. Societal behavior towards people with disability according to the "new generation" sample (Likert scale from 1 to 5, with 1-Not at all in agreement and 5-Strongly in agreement).

Table 4. Societal Positive and negative influence of factors on “the new generation’s” perception of disability (Likert scale from 1 to 5, with 1-No influence and 5-Very great influence).

Table 5. Levers of action and their perceived effectiveness in improving the perception of disability. according to “the new generation” sample. (Likert scale from 1 to 5, with 1-not effective and 5-very effective).

Comments 6: Table 1  should the Mean and SD values contain full stops and not commas?

In all tables, commas have been replaced by full stops.

Comments 7: It would be good to know if survey participants were told to focus on physical forms of disability,intellectual or cognitive disability, or if no instructions were given in terms of interpreting the term “disability”.

In the method we have specified that: “No instructions were given to participants to focus on a specific type of disability, with the aim of obtaining a perception of disability in all its diversity.” (Page 5, line 201)

Comments 8: The results note: On the other hand, characteristics associated with negative perceptions include: aggressiveness, negative attitudes and communication difficulties. These findings highlight the active role that disabled people play in shaping the way they are perceived by non-disabled people and underline the importance of promoting a positive image of their community. This includes an optimistic attitude in the face of challenges, a proactive approach, a desire to connect with others and share their story and individuality by expressing their emotions, whether verbally or non-verbally in the case of communication difficulties. On the other hand, a pessimistic or aggressive attitude on thepart of a disabled person, especially in response to a feeling of rejection by society because of their 'functional diversity', is seen as a negative factor”. I wondered about this idea of the “active role that disabled people play in shaping the way they are perceived” Do we want to suggest that people with disability must be optimistic etc to avoidbeing viewed negatively?

To answer this question, we based ourselves on the article by Hirschauer-Rohmer and Salhani (2002), which confirms the active role that people with disabilities can play in their perception by others: "This type of behavior can promote the emergence of positive affect in the perceiver by familiarizing him or her with the people with disability, thus allowing for more positive interactions [48]" (page 15, line 407). Hence the importance of training people with disability to improve their ability to communicate about themselves and increase their self-confidence and self-image, as indicated on page 15, line 502.

Comments 9: Page 14 refers to the “young people we interviewed”, where it appears only survey methods were used.

We have changed this vocabulary error to "young people who responded to the questionnaire" (page 14, line 475).

Comments 10: I did not see any limitations of the study recognised.

The limitations of the study were added: “Regarding the limitations of this research, it should be noted that all were volunteers to answer the questionnaire (informed consent). We can therefore assume that these people are already aware of, or at least interested in, disability issues. In addition, the fact that the sample was largely composed of women and students may have an impact on generalizability. Finally, the influencing factors and levers for action on the perception of disability highlighted in this article reflect the responses of our sample of young non-experts about disability” (page 15, line 513).

Round 2

Reviewer 3 Report

Comments and Suggestions for Authors

The author/s have done well to address the issues raised in the initial review of this manuscript.

I still question why information about scale construction is provided (Cronbach's alpha), when only descriptive results are presented. Further, in the graphs provided, while percentage and numeric figures are provided for the larger categories, there were no figures visible for the less common categories.

Author Response

Dear Reviewer,

Thank you very much for taking the time to review our research manuscript and for your comments, which have been the basis for the improvement of the study. Please find, as part of this second round of review, the detailed responses below and the corresponding revisions in the re-submitted files.

REVIEWER 3:

Comments 1: I still question why information about scale construction is provided (Cronbach's alpha), when only descriptive results are presented.

Response 1: The information on Cronbach's alphas was removed because, as the reviewer indicated, it was not relevant to the descriptive nature of the results presented (line 334, page 5).

Comments 2: Further, in the graphs provided, while percentage and numeric figures are provided for the larger categories, there were no figures visible for the less common categories.

Response 2: Percentages and numerical values have also been provided for less common categories. (Figure 1 and Figure 2, page 6).